Bridging the TB data gap: in silico extraction of rifampicin-resistant tuberculosis diagnostic test results from whole genome sequence data

http://orcid.org/0000-0003-2049-3116 Ng Kamela C. S. 1 2 kng@itg.be
http://orcid.org/0000-0002-6075-5603 Ngabonziza Jean Claude S. 1 3
Lempens Pauline 1
de Jong Bouke C. 1
http://orcid.org/0000-0002-5490-8968 van Leth Frank 2 4
http://orcid.org/0000-0003-0724-8343 Meehan Conor J. 1 5
1 Mycobacteriology Unit, Department of Biomedical Sciences, Institute of Tropical Medicine , Antwerp , Belgium
2 Amsterdam Institute for Global Health and Development, University of Amsterdam , Amsterdam , The Netherlands
3 Department of Biomedical Services, Rwanda Biomedical Center , Kigali , Rwanda
4 Department of Global Health, Amsterdam UMC, Location Academic Medical Center, University of Amsterdam , Amsterdam , The Netherlands
5 School of Chemistry and Biosciences, University of Bradford , Bradford , UK
Gomez Shawn
Electronic publication date: 2019 Aug 26
Publication date: 2019
Volume: 7
Electronic Location ID: e7564
Received 2019 May 4; Accepted 2019 Jul 29
Copyright: © 2019 Ng et al.
Copyright year: 2019
Copyright holder: Ng et al.
License: This is an open access article distributed under the terms of the Creative Commons Attribution License, which permits unrestricted use, distribution, reproduction and adaptation in any medium and for any purpose provided that it is properly attributed. For attribution, the original author(s), title, publication source (PeerJ) and either DOI or URL of the article must be cited.
License URL: https://creativecommons.org/licenses/by/4.0/

Keywords: Mycobacterium tuberculosis, Rifampicin-resistant tuberculosis, Xpert MTB/RIF, XpertMTB/RIF Ultra, GenoType MDRTBplus v2.0, GenoscholarNTM+MDRTB II, Python, Next generation sequencing, Whole genome sequences, Single nucleotide polymorphism

Funding: Erasmus Mundus Joint Doctorate Fellowship grant 2016-1346 This work was supported by the Erasmus Mundus Joint Doctorate Fellowship grant 2016-1346 to Kamela C. S. Ng. The funders had no role in study design, data collection and analysis, decision to publish, or preparation of the manuscript.

==============================
Background

Mycobacterium tuberculosis rapid diagnostic tests (RDTs) are widely employed in routine laboratories and national surveys for detection of rifampicin-resistant (RR)-TB. However, as next-generation sequencing technologies have become more commonplace in research and surveillance programs, RDTs are being increasingly complemented by whole genome sequencing (WGS). While comparison between RDTs is difficult, all RDT results can be derived from WGS data. This can facilitate continuous analysis of RR-TB burden regardless of the data generation technology employed. By converting WGS to RDT results, we enable comparison of data with different formats and sources particularly for low- and middle-income high TB-burden countries that employ different diagnostic algorithms for drug resistance surveys. This allows national TB control programs (NTPs) and epidemiologists to utilize all available data in the setting for improved RR-TB surveillance.

Methods

We developed the Python-based MycTB Genome to Test (MTBGT) tool that transforms WGS-derived data into laboratory-validated results of the primary RDTs—Xpert MTB/RIF, XpertMTB/RIF Ultra, GenoType MDRTBplus v2.0, and GenoscholarNTM+MDRTB II. The tool was validated through RDT results of RR-TB strains with diverse resistance patterns and geographic origins and applied on routine-derived WGS data.

Results

The MTBGT tool correctly transformed the single nucleotide polymorphism (SNP) data into the RDT results and generated tabulated frequencies of the RDT probes as well as rifampicin-susceptible cases. The tool supplemented the RDT probe reactions output with the RR-conferring mutation based on identified SNPs. The MTBGT tool facilitated continuous analysis of RR-TB and Xpert probe reactions from different platforms and collection periods in Rwanda.

Conclusion

Overall, the MTBGT tool allows low- and middle-income countries to make sense of the increasingly generated WGS in light of the readily available RDT results, and assess whether currently implemented RDTs adequately detect RR-TB in their setting. With its feature to transform WGS to RDT results and facilitate continuous RR-TB data analysis, the MTBGT tool may bridge the gap between and among data from periodic surveys, continuous surveillance, research, and routine tests, and may be integrated within the national information system for use by the NTP and epidemiologists to improve setting-specific RR-TB control. The MTBGT source code and accompanying documentation are available at https://github.com/KamelaNg/MTBGT.

Introduction

Resistance to rifampicin (RIF), the most potent anti-tuberculosis (TB) drug, hampers TB control. RIF-resistant (RR)-TB persists as an urgent public health crisis as only 29% of estimated RR-TB patients worldwide were detected and notified in 2017. Further, 18% of previously treated TB patients were found to have RR-TB (WHO, 2018). The WHO endorsed the following rapid molecular RR-TB diagnostic test (RDTs) to address this concern: Xpert MTB/RIF (Xpert Classic) and the new version Xpert MTB/RIF Ultra (Ultra) (Cepheid, Sunnyvale, CA, USA) which employ heminested real-time polymerase chain reaction and molecular beacon technology (Blakemore et al., 2010); and the line probe assays—GenoType MDRTBplus v2.0 (hereinafter called LPA-Hain) (Hain Lifescience GmbH, Nehren, Germany); and GenoscholarNTM+MDRTB II (hereinafter called LPA-Nipro) (NIPRO Corporation, Osaka, Japan) which rely on multiplex amplification and reverse hybridization of target to both wild-type and mutant probes on a membrane strip (Dheda et al., 2017). These tests were designed to detect RR-conferring mutations within the RIF resistance determining region (RRDR) or hotspot of the rpoB gene (Andre et al., 2017; Blakemore et al., 2010; Ng et al., 2018a, 2018c).

The Xpert assay involves binding of five short overlapping fluorescent probes to wild-type regions of the RRDR. Each of the five probes corresponds to several mutations which inhibit probe binding, thereby disrupting the signal from the respective probe. Thus, when a mutation is detected by Xpert, at least one probe does not bind. Certain mutations which completely interfere with probe binding result in “drop-out” or absent probe, whereas mutations that allow limited probe hybridization are represented by “delayed” probes (Blakemore et al., 2010). Xpert probes are therefore often considered as a proxy for circulating RIF resistance-conferring mutations which may be common or infrequent in a particular country. Understanding their frequency can be crucial for RIF resistance surveillance. For example, mutation Ser450Leu repeatedly detected globally (Coll et al., 2018; Walker et al., 2015) is captured by “absent” Xpert probe E (Ng et al., 2018a).

The implementation of Xpert was a breakthrough in TB diagnosis as it revolutionized the detection of RR-TB worldwide, allowing for prompt identification of patients who need to undergo adapted treatment. Xpert Classic is the most widely deployed RDT globally, implemented as the initial diagnostic tool for all presumptive pulmonary TB patients by 32 out of 48 high TB-burden countries (WHO, 2018). The wide utilization of the tests in both low and high burden TB countries resulted in the production of large volumes of RDT data, although comparisons within and between countries can be difficult due to use of differing technologies.

The utility of next-generation sequencing technologies has been widely studied for improved detection of drug-resistant TB in diverse laboratory settings worldwide (Gardy & Loman, 2018). Next-generation sequencing-based whole genome sequencing (WGS) of Mycobacterium tuberculosis (Mtb) has been shown to accurately detect RR-TB by calling relevant single nucleotide polymorphisms (SNPs) in the rpoB gene (Coll et al., 2018; Miotto et al., 2017). Whole genome sequencing is already being widely implemented in the United Kingdom, the Netherlands, and New York, aimed towards completely replacing phenotypic drug susceptibility testing in the clinic (CRyPTIC Consortium and the 100,000 Genomes Project et al., 2018; de Viedma, 2019). Whole genome sequencing implemented in high burden TB countries was shown to accurately estimate the prevalence of DR-TB (Zignol et al., 2018). The conventional periodic TB drug resistance surveys have been gathering data representative of the Mtb population in poor resource settings, while high-income countries typically apply continuous surveillance (WHO, 2018; Zignol et al., 2018), to help improve the choice of standard TB treatment before full drug sensitivity profile is known (WHO, 2018).

The increasing use of WGS in research and public health initiatives can lead to a disconnect from the RDT-based data being generated routinely in the clinic, and the majority of surveys, widening the Mtb data gap. We aim to bridge this Mtb data gap by transforming WGS data into each of the related RDT data outputs, to facilitate analysis of RR-TB prevalence and underlying mutations regardless of the switch between data generation technologies. This will allow end-users to compare “apples with apples,” and analyze previous historical strains with current isolates representative of the entire TB patient population in the country.

We present the MycTB Genome to Test (MTBGT) tool, a Python 3-based set of scripts that rapidly transform WGS data type into RDT and mutation reports. The modules automatically generate frequencies of RIF-susceptible (RS) and RR-TB samples detected and supplement the RDT probes output with the detected RR-conferring mutation in the format of “wild-type amino acid-codon number-mutant amino acid.”

Materials and Methods

We developed the MTBGT tool, a Python 3-executable that converts Mtb WGS data in the form of variant call format (VCF) or MTBseq (Kohl et al., 2018) tab files into the most likely output that would be observed from Xpert Classic, Xpert Ultra, LPA-Hain, and LPA-Nipro based on previously validated work (Ng et al., 2018a, 2018c). The MTBGT tool can be accessed at https://github.com/KamelaNg/MTBGT.

The MTBGT tool

The MTBGT tool can be run on any python 3-enabled operating system with no additional prerequisites. The MTBGT workflow is shown in Fig. 1. The input is a folder of files derived from an SNP calling pipeline, either in standard raw VCF format or tab format as output from MTBseq. The MTBGT tool assumes the standard H37Rv NC000962.3 genome was used for calling these SNPs. If not, the user may remap the genome positions to the specific RR-TB-related codons using a tab-delimited mapping file. An example of this tab-separated file is bundled with the tool. By default, the module will run all the RDTs on the input files.

Figure 1 The MTBGT tool workflow.

The generated tab-delimited output file includes the Sample name, RIF resistance or susceptibility, the associated mutant codon position and mutation pattern, and a series of 0’s or 1’s indicating absence or presence of the capturing probe for the RDTs, and the RR-conferring mutation (an example output is given in Table 2). A summary table with counts and proportions of detected RS-TB cases, RDT probes, and RR-conferring mutations is also generated (Table S1). These tab-separated output files can be easily imported into Excel as outlined in the associated manual.

Table 1 Rifampicin resistance (RR)-conferring mutations and associated rapid diagnostic test (RDT) results, previously tested and validated*, and used as basis for developing the MTBGT tool.

RR-TB RDT	RR mutation	
Xpert Classic	Xpert Ultra	LPA-Hain	LPA-Nipro	
Capturing probe	Capturing probe	Melting temperature shift	Absent (WT) probe	Developing (MUT) probe	Absent (WT) probe	Developing (MUT) probe	
ND	ND	ND	ND	ND	ND	ND	Val170Phe	
Probe A	rpoB1	3.5	WT1		S1		Ser428Arg	
Probe A	rpoB1	5.9–6.3	WT2		S1		Leu430Pro	
Probe A, Probe B	rpoB1	2.9	WT2		S1		Ser431Gly	
Probe B	rpoB1	3.4	WT2, WT3		S1		Gln432Glu	
Probe B	rpoB2	3.2	WT3		S2		Met434Ile	
Probe B	rpoB2	3.3	WT3		S2		Met434Thr	
Probe B	rpoB1	6.3	WT3		S2		Met434Val	
Probe B	rpoB2	2.8	WT3, WT4		S2		Asp435Glu	
Probe B delayed	rpoB2	5.3	WT3, WT4		S2		Asp435Phe	
Probe B	rpoB2	3.3	WT3, WT4		S2		Asp435Gly	
Probe B	rpoB2	3.5–3.7	WT3, WT4	MUT1	S2	R2	Asp435Val	
Probe B delayed	rpoB2	4.0–4.4	WT3, WT4		S2		Asp435Tyr	
Probe C	rpoB2	6.4	WT4		S2		Asn437Asp	
Probe C	rpoB2; rpoB3	3.0; 2.3	WT5, WT6		S3		Ser441Leu	
Probe C	rpoB2; rpoB3	4.7; 2.3	WT5, WT6		S3		Ser441Gln	
Probe D	rpoB3	3.7–3.9	WT7	MUT2B	S4	R4b	His445Asp	
Probe D	rpoB3	4.9	WT7		S4		His445Gly	
Probe D	rpoB3	3.5–3.6	WT7		S4		His445Leu	
Probe D	rpoB3	3.4–3.5	WT7		S4		His445Asn	
Probe D	rpoB3	3.6	WT7		S4		His445Gln (CAG)	
Probe D	rpoB3	4.1	WT7		S4		His445Gln (CAA)	
Probe D delayed	rpoB3	1.9	WT7		S4		His445Arg	
Probe D	rpoB3	4.7	WT7		S4		His445Ser	
Probe D	rpoB3	4.9	WT7		S4		His445Thr	
Probe D	rpoB3	3.2–3.3	WT7	MUT2A	S4	R4a	His445Tyr	
Probe D	rpoB4B	5.0	WT7		S4		Lys446Gln	
Probe E	rpoB3	4.0	WT8		S5		Ser450Phe	
Probe E	rpoB3; rpoB4A	2.5–2.9; 6.0–6.5	WT8	MUT3	S5	R5	Ser450Leu	
Probe E	rpoB3; rpoB4A	2.3–2.7; 3.3–3.7	WT8		S5		Ser450Trp	
Probe E delayed	rpoB4B	5.7–6.1	WT8		S5		Leu452Pro	
ND	ND	ND	ND	ND	ND	ND	Ile491Phe	
Notes

* (Ng et al., 2018a, 2018b); Capturing probe, RDT probe associated with the RR-conferring mutation; Melting temperature shift, difference between mutant (MUT) and wild-type (WT) melting temperatures; ND, ‘not detected’, refers to mutations outside the RR determining region, not detected by the RR-TB RDTs.

Validation and sample application of the MTBGT modules

We simulated VCF files for internal validation of the MTBGT modules. We then randomly chose a VCF file and edited it in Notepad++ 7.6.3 (https://notepad-plus-plus.org/) to contain all previously tested and validated RR-conferring mutations (Andre et al., 2017; Miotto et al., 2017) which were mapped to known RDT results (Ng et al., 2018a, 2018c) (Table 1). We generated files with single and multiple RR mutations to ensure the tool is robust for all scenarios. These simulated VCFs are provided with the tool.

Table 2 Example of combined results from the MTBGT tool rapid diagnostic test (RDT) modules supplemented by the rifampicin resistance (RR)-conferring mutations detected.

Filename	RIF resistance	Codon number	Codon	Xpert Classic	Xpert Ultra	LPA-Hain	LPA-Nipro	RR mutation	
Capturing probe	Probe pattern	Capturing probe, melting temperature shift	Probe pattern	Capturing probe	Probe pattern	Capturing probe	Probe pattern	
DRC-052577	Detected	450	TTG	Probe E	1 1 1 1 0 1	rpoB3,rpoB4A; 2.5–2.9, 6.0–6.5	1 1 0 0 1	WT8, MUT3	1 1 1 1 1 1 1 0 0 0 0 1	S5, R5	1 1 1 1 0 0 0 0 1	Ser450Leu	
DRC-091003	Detected	452	CCG	Probe E delayed	1 1 1 1 1 0	rpoB4B; 5.7–6.1	1 1 1 1 0	WT8	1 1 1 1 1 1 1 0 0 0 0 0	S5	1 1 1 1 0 0 0 0 0	Leu452Pro	
1993-09004	Detected (only by rpoB Sanger sequencing module)	170	TTC	Not detected	Not detected	Not detected	Not detected	Val170Phe	
DRC-101308	Detected (only by rpoB Sanger sequencing module)	491	TTC	Not detected	Not detected	Not detected	Not detected	Ile491Phe	

To show real-world applicability, the MTBGT tool was run on WGS of RS and RR-TB WHO Tropical Disease Research (TDR) strains with diverse resistance patterns and geographic origins stored in the Belgian Coordinated Collections of Microorganisms in the Institute of Tropical Medicine including the 47 TDR-TB strains tested in the previous validation of the RDTs against the available rpoB Sanger sequences of the strains (Ng et al., 2018a, 2018c; Vincent et al., 2012). The fastQ files (ENA accession PRJEB31023) for these samples were run through the MTBseq pipeline with default settings (Kohl et al., 2018) to generate the tab files for input to MTBGT. Additionally, WGS from 324 phenotypically RR-TB isolates of retreatment TB patients in Kinshasa, the Democratic Republic of Congo (DRC) collected in 2005–2010 (Meehan et al., 2018), and 233 WGS isolated 1991–2010 from Rwanda were subjected to the MTBseq pipeline and run through the MTBGT tool.

We were then able to extract the associated Xpert results from the 1991 to 2010 Rwandan dataset and compare it with actual Xpert results from 2012 to 2017 (Ng et al., 2018b).

Results

We tested the MTBGT tool on a 64-bit Windows 10 Enterprise computer with a 2.50 GHz processor and a 8.00 GB of RAM. The running time was 39 milliseconds for one MTBseq tab file and 530 milliseconds for a VCF file.

The MTBGT tool correctly transformed the WGS-derived SNPs in the VCF and MTBseq tab files into the laboratory-validated RDT probe reactions (Table 2), and accurately detected all previously validated RR-conferring mutations in the simulated VCF and MTBseq clinical WGS data from the TDR, DRC (Figs. 2A and 2B), and Rwanda (Figs. 3A and 3B) strains, including double nucleotide changes—two SNPs covering two loci or genome positions—such as the CAC → AGC and CAC → TCC His445Ser mutations and combinations of any two unlinked mutations. The generated frequency and proportion tables (Table S1) showed the distribution of the RS and RR-TB strains, the RDT probe reactions, and the RR-conferring mutations detected.

Figure 2 Distribution of (A) rifampicin-sensitive samples and Xpert Classic probes, and (B) rifampicin resistance-conferring mutations among rifampicin-resistant tuberculosis isolates in Kinshasa, DRC from 2005 to 2010, detected by the MTBGT tool.

Figure 3 Distribution of (A) rifampicin-sensitive samples and Xpert Classic probes, and (B) rifampicin resistance-conferring mutations among rifampicin-resistant tuberculosis isolates in Rwanda from 1991 to 2010, determined by the MTBGT tool and (C) distribution of r.

The in silico extracted Xpert results from the Rwandan WGS collected in 1991–2010 revealed that the majority of RR-TB cases detected were linked with Xpert Classic probe E (Fig. 3A). This observation was consistent with the actual Xpert results gathered in 2012–2017. Figure 3C shows the distribution of 12 years worth of Xpert results in Rwanda. The MTBGT tool facilitated this continuous analysis of in silico predicted Xpert results from 2005 to 2010 WGS and actual Xpert results in 2012–2017, and allowed us to plot the distribution of absent probe E against the notified RR-TB cases from surveillance programs and national surveys in Rwanda. The low WGS sampling reflected in Fig. 3C is explained by the limited TB cultures kept in the freezer in 2005–2010. Notably, we also observed the divergence of documented probe reactions from the predominant absent probe E in 2013–2017.

Xpert Classic probe E was also predominantly seen in samples from Kinshasa, DRC (Fig. 2A; Table S1). Mutation Ser450Leu, linked with Xpert Classic probe E, was the major RR-conferring mutation observed in both settings (Figs. 2B and 3B).

Discussion

To make sense of the increasingly produced WGS in light of the existing and readily available routine RDT results, we developed, validated, and applied the MTBGT, a Python-implemented tool that extracts RR-TB RDT results and RR-conferring mutations from WGS-derived SNP data.

If a country performed prior drug resistance surveys with a different diagnostic algorithm, the MTBGT tool allows for comparing data of different formats and sources, facilitating analysis of previous and current RR-TB case counts, probe reactions, and mutations, such as periodic DRS results conducted as cross-sectional surveys, often with different technology from the previous one. Although WGS data give a much higher resolution than the RDTs, low- and middle-income high RR-TB burden countries are not yet capable of implementing routine WGS for all presumptive TB patients due to limitations in funding and logistic challenges (Meehan et al., 2019). However, RDTs are used routinely in such countries, producing a wealth of RR-TB data, whereas WGS is used for regular country-wide drug resistance surveys by the WHO (Zignol et al., 2018), and implemented in low burden high-income countries. But since RDT results cannot be upscaled to WGS results, for comparison of these two data sources, WGS must be downscaled to the RDT results. By downscaling, the MTBGT tool adds to a substantial history of RDT probe data which amplifies surveillance analyses that can inform national TB control programs (NTPs) of low- and middle-income settings. This is similar to how CRISPR-based strain typing of M. tuberculosis (termed spoligotyping) can be predicted from WGS reads through SpoTyping (Xia, Teo & Ong, 2016). Although spoligotyping is known to have a lower resolution than WGS for typing, it is still widely used in many low- and middle-income TB endemic countries due to its low cost (Montoya et al., 2013; Suzana et al., 2017; Tulu & Ameni, 2018). We foresee our tool being used in a similar manner, until WGS capabilities become more commonplace in low- and middle-income high RR-TB burden countries.

We show in Fig. 3C how the MTBGT tool would be beneficial in Rwanda, a low-income country which uses Xpert for routine diagnostics and WGS for WHO-led 5-year drug resistance surveys and selected research projects. Figure 3C reflects continuous distribution of RR-TB cases and Xpert absent probe E in Rwanda for more than a decade. These trends of RR-TB and absent probe E boost surveillance analyses that can inform the Rwandan NTP and support public health efforts for improved RR-TB control. The MTBGT tool allows for such comparisons and continuous distributions to be made, enabling low- and middle-income researchers to utilize all of their data for surveillance of RR-TB. This analysis of Rwandan datasets is a proof-of-concept that the MTBGT tool provides an opportunity to compare and analyze old and new data produced by different technologies bridging the RR-TB data gap. Care must be taken that the comparison is valid given the populations sampled. The predominant Xpert Classic probe E observed in Rwanda is supported by the associated RR mutation Ser450Leu (Tables 1 and 2) being present in the primary circulating multidrug-resistant TB clone in this setting (Ngabonziza et al., 2018), and also most frequently associated with global RR-TB (Cohen et al., 2015; Coll et al., 2018; Georghiou et al., 2016).

Programmatic surveillance is possible with only the RIF susceptibility status of patients’ samples, but at a different level. The RDT probe information which represents the underlying RR-conferring mutation provides an added critical value for epidemiological studies and surveillance (Ng et al., 2018b). This is exemplified by Xpert probe B and probe binding delay in South Africa being associated with false RR-TB results (Berhanu et al., 2019). In this setting, patients who were true RS but diagnosed as RR were treated with less effective and more toxic multidrug-resistant TB drugs relative to the standard TB treatment. Further, non-routine WGS alone may give incorrect distribution of circulating mutations in the setting due to culture and sampling biases, as shown by the limited stored cultures resulting in low WGS sampling in Rwanda in 2005–2010.

Settings which employ different RDTs can refer to Table 1 where they can match laboratory-validated Xpert Classic and Xpert Ultra probe signatures with a specific RR-TB mutation type (Ng et al., 2018a, 2018c). This would be helpful for settings such as South Africa which implements Xpert Classic and concurrently transitions to Xpert Ultra (Berhanu et al., 2018), and many other countries that will shift to Xpert Ultra in the near future, such as Rwanda by 2020.

The genome-based approach of the MTBGT tool also allows for reporting of disputed mutations that confer occult RR and are frequently missed by the Mycobacterium Growth Indicator Tube phenotypic DST (Ng et al., 2018a; Van Deun et al., 2015). For instance, disputed mutation Leu452Pro, epidemiologically linked with an extensively drug-resistant TB outbreak in KwaZulu-Natal, South Africa in 2005 (Cohen et al., 2015; Ioerger et al., 2009), was detected in some strains from Kinshasa, DRC, and Rwanda (Figs. 2B and 3B; Supplemental File). Mutation Leu452Pro is captured by delayed Xpert Classic probe E, denoting partially inhibited probe E fluorescence (Lawn & Nicol, 2011; Ng et al., 2018a), and specifically identified by the unique combination of Xpert Ultra probe rpoB4B and corresponding melting temperature shift (Table 1), provided sufficient Mtb DNA is detected. Further, mutation Leu452Pro was reported to be missed in clinical samples by LPA-Hain due to its end-probe location (Al-Mutairi, Ahmad & Mokaddas, 2011; Rigouts et al., 2013), thus contributing to the RR-TB detection gap. The ability of the MTBGT tool to rapidly and accurately detect disputed mutations is therefore important.

As a supplementary feature, the MTBGT tool picks up any RR-conferring mutation present in the VCF or MTBseq tab file, and may help assess whether RDTs sufficiently detect RR-TB cases in specific settings.

Looking into the future when WGS will be implemented as the primary diagnostic tool for all presumptive TB patients to confirm TB and/or RR-TB, the MTBGT tool will bridge the TB data gap by allowing comparison of past Xpert and LPA results with current in silico predicted results from routine whole genome sequences. These data would be comparable based on a common statistical parameter and representative of the presumptive TB population in the setting. With more whole genomes of M. tuberculosis sequenced, the in silico predicted probe distribution will more accurately capture the circulation of RR-TB mutations in the setting. The tool can do so by precisely revealing the divergence of current probes from a previously documented predominant probe over time, implicative of other circulating mutations such as that shown by actual Xpert results in 2013–2017 (Fig. 3C).

The MTBGT tool was developed to aid surveillance efforts of low- and middle-income high RR-TB burden countries with likely fragmented data collection systems (Mazumdar, Satyanarayana & Pai, 2019). Through facilitated continuous analysis of previous historical and current RR-TB data, as well as clinical data from large aggregated files and routine laboratory and periodic survey RDT results combined with WGS-derived data from national surveillance programs and public data repositories, the MTBGT tool may create a larger picture of the RS-TB and RR-TB burden in a country. The RS and RR-TB counts and proportions and RDT probe reactions generated by the MTBGT tool may contribute to an extensive database with years’ worth of data for continuous statistical modeling analyses and surveillance investigations.

Potentially, the MTBGT tool may be integrated in the national TB diagnostic algorithm through the existing information system and connectivity platform or the newly implemented WHO cloud-based software (Dean, 2019). The prospective application of the MTBGT tool may bridge and transform the Mtb data gap into action points for RR-TB clinicians to provide appropriate care for the individual TB/RR-TB patient, and the NTP, public health officials, and policy makers to intervene at the population-level for improved and sustained RR-TB control (Gardy & Loman, 2018). The MTBGT modules could also be expanded to report non-RRDR RR-conferring mutations and include other TB drugs—e.g. isoniazid, pyrazinamide, and fluoroquinolones.

Conclusions

The MTBGT tool leverages improved access to next-generation sequencing technologies in this genomic epidemiology era of TB, and complements Mtb WGS by rapidly transforming WGS files that store genomic sequence variations to validated outputs of the RR-TB RDTs. The prospective application of the MTBGT tool within a nationwide genomic epidemiology program will bridge the Mtb data gap among routine RDT data, research setting WGS, and periodic survey and continuous surveillance rpoB sequencing and WGS data. This may result in facilitated continuous analysis of the circulating RDT probes and underlying distribution of RR-conferring mutations in the setting, and may help assess whether currently implemented RDT(s) serve the detection of RR-TB cases in the country.

Supplemental Information

Supplemental Information 1 Summary tables of counts and proportions of rifampicin-susceptible TB samples, rapid diagnostic test probes, and rifampicin-resistant TB mutations detected by the TBGT tool from the 2005 to 2010 isolates in Kinshasa, DRC.

Click here for additional data file.

We thank Button Ricarte for his insightful feedback on the manuscript.

Additional Information and Declarations

Competing Interests

Author Contributions

Data Availability

The authors declare that they have no competing interests.

Kamela C. S. Ng conceived and designed the experiments, performed the experiments, analyzed the data, prepared figures and/or tables, authored or reviewed drafts of the paper, approved the final draft.

Jean Claude S. Ngabonziza contributed reagents/materials/analysis tools, approved the final draft.

Pauline Lempens approved the final draft, curated the TDR WGS.

Bouke C. de Jong approved the final draft.

Frank van Leth approved the final draft.

Conor J. Meehan conceived and designed the experiments, performed the experiments, contributed reagents/materials/analysis tools, authored or reviewed drafts of the paper, approved the final draft.

The following information was supplied regarding data availability:

All scripts and data for validation, the TBGT source code and accompanying documentation are available at https://github.com/KamelaNg/MTBGT, with the open source license at https://github.com/KamelaNg/MTBGT/blob/master/LICENSE.

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
