# Peer review of "Bridging the TB data gap: in silico extraction of rifampicin-resistant tuberculosis diagnostic test results from whole genome sequence data"

_PeerJ, doi:10.7717/peerj.7564_

## Round 0.1 · original submission · Major Revisions

While the reviewers found the manuscript of interest, several concerns were raised with regard to the validity and/or strength of the conclusions. Please address these reviewer concerns.

Reviewer 1 ·

Basic reporting

Text is written in an understandable way with only minor mistakes, such as word doublings (line 172), missing punctuation marks (line 144 - 147), etc.
Figure legends names goes a little confused and i would have chosen other figures, which fit better to the results. Tables are well labeled and described, however, I would cite the paper in legend for table 1.
Introduction and background quite long compared to results part, however, I miss a part how RDT's work in general.

Experimental design

The study is well designed and the program is first trained with in-silico data and afterwards checked with "real" data.
The method part is written in detail and easy to understand so that the analyses can be reproduced. In my eyes, however, the results do not reflect the aim of the study. In my opinion the results support the fact that the high resolution of the NGS data should not be broken down to RDT. For comparisons of different years it would be enough to compare resistant status R/S, so that even different GeneXpert results could be compared. Line 86 says "comparison within and between countries can be difficult due to use of differing technologies", but this is still the case when one country uses Xpert classic and the other uses Xpert ultra, isn't it?

Validity of the findings

For me the validity of the study is not clear. It is stated that this tudy should close a knwoledge gap, which I don't see. Of course it makes sense to make different technologies comparable, but I see no reason to break down such a high-resolution technology as NGS to such a primitive technology as RDT. The author stated that this will make surveillance easier (line 231) or allowing for longitudinal analysis of RR-TB prevalence, but in my mind for surveillance and prevalence estimation a status such as R for resistant or S for susceptible would be enough. The knowledge of the specific probe is not a knowledge gain but loss of information compared to a specific SNP you received from the sequencing. Especially the last sentence of the clonclusion is missleading. Noone would take the RR-TB results as marker for transmission. Even the results here show, that there are a few SNP prominently circulating but to get an idea of transmission one would genotype the strains and when WGS data is available it would be possible to do a sufisticated contact tracing by SNP distances.

Additional comments

How you decided on the name of the tool? When I searched for TBGT in github there were 5 hits and two other software were named exactly the same with small letters. Google had several hits but on the first pages your TBGT was not found at all. Additionally I don't get the connection of transforming NGS data to RDT data and test a TB Genome. That is why I would think of a new name for the tool.
I am not so sure, who needs such a tool. I liked the previous publication about the correlation of RDTs and SNPs, but in my mind a cool story would be the transformation the other way round, although I know, that this is not possible with Xpert Classic results for example.

In the supplementary tables I would always put Rif susceptible on top or on the bottom of the table to make them comparable.

Reviewer 2 ·

Basic reporting

The author have developed a useful tool that transforms SNP results to RDT format. This will be helpful to public health systems (National Health Control Programmes) in several countries that utilize the relevant technologies in their programmes. The study is aimed at converging all data(past and present) n the public domain to offer offer the most effective therapies to their patients.
The language used in the paper is professional and has clarity.However the repetitive word "all" in line 172 needs to be removed. Authors should explain the significance of the results in relation to public health initiatives so that it is easier to understand the function of this tool.
Contemporary references have been quoted to illustrate the background
however the figures 2a, 2b,3a, 3b do not describe the units of the y-axis

Experimental design

The article meets the aims of the Journal and identifies a critical gap in the utility of genomic data generated at different levels. The TBGT tool is an attempt to converge this data in a more powerful format for betterment of TB treatment and control. This utility needs to be described better to the reader for easier comprehension.
Please mention in Methods (line 131) whether vcf file is araw file or a filtered vcf file. If the latter, then the parameters used to filter the file shouold be mentioned.

In the TGBT, the output is in the form of a text format. It would be helpful if the result format was in excel as well to ease comparison of data

Validity of the findings

The utility of this approach to capture data from multiple data repository sites would be useful to mention. The approach to undertake this could be mentioned in the discussion especially for countries like India where consolidation of data is likely to be fragmented.

---

## Round 0.2 · accepted · Accept

Thank you for addressing the reviewer concerns. I think providing additional information as to the practical constraints of developing countries greatly helps in providing an appropriate context for why this tool has value. Congratulations again.